

# Phytotoxin synthesis genes and type III effector genes of *Pseudomonas syringae* pv. *actinidiae* biovar 6 are regulated by culture conditions

Karin Hirose[1,2], Yasuhiro Ishiga[2] and Takashi Fujikawa[1]

[1] Institute of Fruit Tree and Tea Science, National Agriculture and Food Research Organization (NARO), Tsukuba, Ibaraki, Japan
[2] Faculty of Life and Environmental Science, University of Tsukuba, Tsukuba, Ibaraki, Japan

## ABSTRACT

The kiwifruit bacterial canker (*Pseudomonas syringae* pv. *actinidiae*; Psa) causes severe damage to kiwifruit production worldwide. Psa biovar 6 (Psa6), which was isolated in Japan in 2015, produces two types of phytotoxins: coronatine and phaseolotoxin. To elucidate the unique virulence of Psa6, we performed transcriptomic analysis of phytotoxin synthesis genes and type III effector genes in *in vitro* cultivation using various media. The genes related to phytotoxin synthesis and effectors of Psa6 were strictly regulated in the coronatine-inducing mediums (HS and HSC); 14 of 23 effector genes and a *hrpL* sigma factor gene were induced at 3 h after transferring to the media (early-inducible genes), and phytotoxin synthesis genes such as *argD* of phaseolotoxin and *cfl* of coronatine were induced at 6 and 12 h after transferring to the media (late-inducible genes). In contrast, induction of these genes was not observed in the *hrp*-inducing medium. Next, to examine whether the changes in gene expression in different media is specific to Psa6, we investigated gene expression in other related bacteria. For Psa biovar 1 (Psa1), biovar 3 (Psa3), and *P. s.* pv. *glycinea* (Psg), no clear trends were observed in expression behavior across various culture media and incubation times. Therefore, Psa6 seems to exert its virulence efficiently by using two phytotoxins and effectors according to environmental changes. This is not seen in other biovars and pathovars, so it is thought that Psa6 has acquired its own balance of virulence.

# INTRODUCTION

Kiwifruit belong to the family *Actinidiaceae* and the genus *Actinidia*. They include *Actinidia deliciosa*, a species whose flesh is green, and *A. chinensis*, a species whose flesh is yellow or red. In Japan, kiwifruit seedlings were first introduced from New Zealand in the 1970s and are now popular as edible fruit (*Ushiyama, 1993*). However, around 1980, kiwifruit bacterial canker broke out and caused serious damage to kiwifruit production (*Takikawa, Serizawa & Ichikawa, 1989*). Since then, this disease has resulted in economic and productive damage in kiwifruit orchards throughout Japan.

Corresponding author
Takashi Fujikawa, ftakashi@affrc.go.jp

The pathogen that causes kiwifruit bacterial canker is *Pseudomonas syringae* pv. *actinidiae* (Psa). It has been further classified into biovar 1, 2, 3, 5, and 6, reflecting its physiological and genetic diversity (*Chapman et al., 2012*; *Fujikawa & Sawada, 2016*; *Fujikawa & Sawada, 2019*; *Sawada & Fujikawa, 2019a*; *Sawada & Fujikawa, 2019b*). In Japan, Psa is highly diverse, with significant differences in the types or presence of phytotoxins produced among biovars (*Sawada & Fujikawa, 2019a*; *Sawada & Fujikawa, 2019b*). In general, the plant pathogenic bacteria group, *P. syringae,* is highly differentiated in pathogenicity, and it is known to have more than 60 different pathovars with different host plants and symptoms (*Sawada, 2014*). However, of all known *P. syringae* pathogens, only Psa biovar 6 (Psa6) has been reported to produce two phytotoxins (coronatine and phaseolotoxin) simultaneously. Phytotoxins are known to be important factors in the virulence of plant pathogens because they have a significant effect on plants, even at low concentrations (*Bender, 1999*; *Strange, 2007*). Coronatine, a phytotoxin produced by Psa6, is structurally similar to the plant hormone jasmonic acid (*Geng et al., 2014*). It has been reported to inhibit host defense responses by disrupting hormonal signals and reopening closed stomata (*Geng et al., 2014*). Phaseolotoxin, another phytotoxin produced by Psa6, has been shown to induce halos around lesions in leaves by inhibiting the function of ornithine carbamoyltransferase, which is an enzyme involved in arginine biosynthesis in host plants (*Mitchell, 1976*; *Patil et al., 1976*; *Tamura et al., 2002*). These phytotoxins have been extensively studied, and the conditions under which they are induced have been clarified in some bacteria. For example, *P. syringae* pv. *glycinea* (Psg), which is a causal pathogen of soybean (*Glycine max*), produces coronatine when cultured in HS and HSC medium (*Hoitink & Sinden, 1970*; *Palmer & Bender, 1993*), and *P. syringae* pv. *phaseolicola*, which is a causal pathogen of the common bean (*Phaseolus vulgaris*), produces the highest amounts of phaseolotoxin at 18 °C (*Nüske & Fritsche, 1989*).

In addition to the two phytotoxins, Psa6 produces other virulence factors, type III effector proteins. Effectors are known to suppress or interfere with the plant's defense response, increasing the pathogen's virulence (*Jones & Dangl, 2006*). The type III secretion system is a specialized molecular machine of gram-negative bacteria that allows the effector proteins to be injected directly into the plant host cells (*Coombes, 2009*). Effector genes of Psa6 in the early stages of infection can be classified into several groups based on differences in expression patterns (*Fujikawa & Sawada, 2019*). Although the core genome sequences of Psa are almost the same among biovars, the presence and combination of phytotoxins and effector genes are diverse. Therefore, the diversity of effectors and phytotoxins may be due to accessory genomes encoded by transposons and other mobile genetic elements (*Sawada & Fujikawa, 2019a*; *Sawada & Fujikawa, 2019b*).

It is important to understand the survival strategies, evolution, and natural adaptations of Psa. In particular, determining the expression behavior of the two phytotoxin synthesis genes and effector genes of Psa6 can provide important information. We used RNA sequencing (RNA-seq) and reverse transcription-quantitative PCR (RT-qPCR), to perform transcriptomic analysis of these genes when cultured under phytotoxin-inducing or effector-inducing conditions.

## MATERIALS & METHODS

### Bacterial strains, media, and culture conditions

Psa6 MAFF 212134, Psa1 MAFF 302145, Psa3 MAFF 212104, and Psg MAFF 301683 were used as the representative strains. These strains can be obtained from NARO Genebank (https://www.gene.affrc.go.jp/index_en.php). For the pre-culture, each strain was scraped from LB (*Bertani, 1951*) solid medium and was transferred to new LB liquid media. The cultures were then shaken overnight. After shaking, the pre-cultures were centrifuged at 6,000×g for 5 m, then each pellet was resuspended with water to OD (660 nm) = 0.6. Two milliliters of resuspension were inoculated into 15 mL of respective test media and the cultures were shaken at 160 rpm at 27 or 18 °C. As test media, LB medium, HS medium, HSC medium, and *hrp*-inducing medium were used (Table 1). HS and HSC media have been reported to activate coronatine production of Psg (*Hoitink & Sinden, 1970*; *Palmer & Bender, 1993*). On the other hand, *hrp*-inducing medium has been reported to induce the expression of *hrp* genes and effector genes of Psg (*Huynh, Dahlbeck & Staskawicz, 1989*). Additionally, it has been reported that cultures at 18 °C, compared to 27 °C, induce the production of phytotoxins such as coronatine and phaseolotoxin (*Nüske & Fritsche, 1989*; *Palmer & Bender, 1993*). After incubation at 0, 3, 6, and 12 h, one mL of each culture was taken and used for extraction of DNA or RNA.

### DNA and RNA extraction

DNA was extracted using the DNeasy mini kit (QIAGEN, Hilden, Germany) and RNA was extracted using the RNeasy mini kit (QIAGEN). DNA was used to confirm primer design, and RNA was used for RNA-Seq and RT-qPCR.

### RNA-Seq

RNA-Seq was performed according to our previous study (*Fujikawa & Sawada, 2019*). Briefly, total RNA was extracted from the 3 h culture in HS or HSC media at 27 °C or 18 °C, were treated with the RiboMinus Transcriptome Isolation Kit for bacteria (Thermo Fisher Scientific Inc., Waltham, MA, USA), and rRNAs were almost depleted from the total RNAs. The RNA-Seq library was constructed from sample RNAs using the Ion Total RNA-Seq Kit v2 (Thermo Fisher Scientific Inc.) and Ion Xpress RNA-Seq barcode (Thermo Fisher Scientific Inc.). Subsequently, the RNA-Seq templates were prepared using the Ion PGM Hi-Q View OT2 Kit (Thermo Fisher Scientific Inc.) on an Ion OneTouch 2 system. The templates were sequenced with an Ion PGM Hi-Q View Sequencing Kit (Thermo Fisher Scientific Inc.) and a 318 Chip Kit v2 (Thermo Fisher Scientific Inc.) on an Ion PGM next generation sequencer. Sequence data were analyzed as RNA reads using the CLC Genomics Workbench ver. 11 (QIAGEN) and the statistical software *R* ver. 3.5.1 with various packages (*edgeR*, *limma*, and *gplots*) according to a previous report (*Fujikawa & Sawada, 2019*). Namely, using high-quality sequence reads with the reference genome of Psa6 MAFF 212134 (accession no. MSBW00000000), the reads per kilobase of exon per million mapped reads (RPKM) values were obtained and normalized. Then, using *R* software, the gene expression values relative to the 27 °C HS medium were calculated to

**Table 1  Each composition of media used in this study.**

| Per liter of medium | HS | HSC | *hrp*-inducing | LB |
|---|---|---|---|---|
| NH$_4$Cl | 1.0 g | 1.0 g | | |
| (NH$_4$)$_2$SO$_4$ | | | 1.0 g | |
| MgCl$_2$.6H$_2$O | | | 0.34 g | |
| MgSO$_4$.7H$_2$O | 0.3 g | 0.2 g | | |
| KH$_2$PO$_4$ | 4.1 g | 4.1 g | 6.8 g | |
| K$_2$HPO$_4$ | 3.6 g | 3.6 g | | |
| KNO$_3$ | | 0.3 g | | |
| NaCl | | | 0.1 g | 10 g |
| D-glucose | 10 g | 20 g | | |
| D-fructose | | | 1.8 g | |
| FeCl$_3$.6H2O | 2 μM | 20 μM | 20 μM | |
| Yeast extract | | | | 5.0 g |
| Bactopeptone | | | | 10 g |

log$_2$ values of fold change (log$_2$FC) with FDR (false discovery rate) values and each effect size (eta squared) values under the four culture conditions.

For the RT-qPCR analysis, reference gene candidates were selected according to the following conditions: (1) low FC (fold change) value, which means that expression is not strongly affected by culture conditions (log$_2$FC= $\leq \pm$ 0.2), and (2) high and stable CPM (counts per million) value (sum of log$_2$CPM). When genes meeting the above conditions were found, primers were designed and used for RT-qPCR analysis.

## RT-qPCR

For RT-qPCR analysis, cDNA was synthesized from total RNA of each culture sample using the PrimeScript II 1st strand cDNA Synthesis Kit (Takara Bio, Shiga, Japan) with random hexamers to yield a total volume of 20 μL. Then, 180 μL of nuclease-free water was added to bring the total volume to 200 μL. The synthesized cDNA was subjected to RT-qPCR using TB Green Premix Ex Taq II (Takara Bio). The PCR conditions were set as follows: denaturation at 96 °C for 10 m; a cycle of 96 °C for 15 s, 58 °C for 30 s, and 72 °C for 1 m, repeated 40 times; then 96 °C for 15 s, 58 °C for 1 m, and 95 °C for 15 s to form a melting curve. DNA of Psa6 was used as a positive control and nuclease-free water was used as a negative control. Then, based on the results of RNA-Seq (described above) we selected the genes that had similar expression levels across all culture conditions as reference genes for each bacterium. According to the method of *Smith, Lovelace & Kvitko (2018)*, RT-qPCR results were processed using the following equation:

$$NRQ = \frac{E_{GOI}^{-(Ct_{GOI\ in\ sample} - Ct_{GOI\ in0h\ sample})}}{\sqrt[n]{\prod_{k=1}^{n} E_{RG}^{-(Ct_{RG\ in\ sample} - Ct_{RG\ in0h\ sample})}}} \qquad (1)$$

where NRQ is the normalized relative quantity, $E$ is amplification efficiency of qPCR, GOI is gene of interest, and RG is reference gene

To solve the NRQ ($\Delta \Delta$Ct value), a geometric mean was calculated from the RT-qPCR results of three reference genes as shown by the denominator equation, and $\Delta$Ct of each
sample was calculated based on it. Then, Δ ΔCt for each condition was calculated on the basis of the RT-qPCR result for 0 h inoculation under each condition. Moreover, the results of RT-qPCR analysis were clustered using the unweighted pair group method with arithmetic mean (UPGMA) and output to heat maps using *R* (ver. 3.5.1) software with two packages (*Heatplus*, and *viridis*).

## PCR primers

Primers were designed and tested for the following: coronatine synthesis-related genes (*cfl*, *cfa1*, *cfa5*, *cfa9*, *cmaD*, *cmaU*, *corR*, *corS*, and *corP*); phaseolotoxin synthesis-related genes (*argK* and *argD*); and type III effector and related genes (*avrD1*, *avrE1*, *avrpto5*, *avrRpm1*, *avrRps4*, *hopAA1-1*, *hopAE1*, *hopAH1*, *hopAI1*, *hopAS1*, *hopAU1*, *hopAZ1*, *hopD1*, *hopJ*, *hopM1*, *hopN*, *hopQ1*, *hopR1*, *hopS2*, *hopY1*, *hopZ3*, *hrpK*, and *hrpL*) (Table S1). The primers for the five selected reference genes (details above) were also designed. These were "50S ribosomal protein L13", "bifunctional glutamine synthetase adenylyltransferase/deadenyltransferase", "tRNA dihydrouridine (20/20a) synthase *dusA*", "transcriptional regulator *ftrA*", and "microcin ABC transporter ATP-binding protein". It was confirmed that these primer sets can be applied not only to Psa6 but also to Psa1 (excluding the coronatine gene) and Psa3 (excluding the two phytotoxin genes) equally (Fig. S1). The similar primer sets were also designed for the Psg homolog genes (Table S1).

To confirm the validity of the designed primers, the extracted DNA was subjected to PCR using Takara Ex Taq Hot Start Version (Takara Bio) with the following conditions: denaturation at 95 °C for 10 min; a cycle of 95 °C for 30 s, 65 °C for 30 s, and 72 °C for 1 m, repeated 40 times; and extension at 72 °C for 7 m. It was confirmed that respective amplicons were single bands by electrophoresis (Fig. S1), and each melt peak of single band using melt curve analysis was also measured (Table S1). In addition, each amplification efficiency of primer sets was calculated (Table S1). These appropriate primer sets were used for RT-qPCR analysis.

## RESULTS AND DISCUSSION

### Gene expression of Psa6 under different culture conditions using RNA-Seq analysis

RNA-Seq data of Psa6 was archived in NCBI GEO (accession no. GSE149743). RNA-Seq of Psa6 revealed the behavior of 5,796 genes (Table S2). The expression values from the HS medium at 27 °C were used to calculate the expression values from the HS medium at 18 °C and the HSC medium at 27 and 18 °C. Seventy-eight genes had an FDR value lower than 0.1 (Table 1). In contrast to the HS medium at 27 °C, 26 genes were induced in the HSC medium at 18 °C, including the type III chaperone ShcN protein gene and the type III effector (translocator) *hrpK*. In particular, *hrpK* had a large effect size (0.96). However, there was no significant induction of other type III effector genes. In addition, neither the induction nor the suppression of genes involved in toxin production was observed. In contrast to HS medium at 27 °C, 17 genes were suppressed in both the HSC medium and the 18 °C condition; however, no genes thought to be involved in virulence were significant (Table 2). Similarly, no virulence genes were significantly induced or suppressed under the

different culture conditions. The HSC medium is an improvement of the HS medium and induces coronatine production (*Palmer & Bender, 1993*). However, we found no significant change in the expression of phytotoxin genes. This suggests limited coronatine synthesis in the 3 h incubation of Psa6. Incubation of *P. syringae* at 18 °C can induce phytotoxin production compared with incubation at 27 °C (*Nüske & Fritsche, 1989*; *Palmer & Bender, 1993*), but we did not observe significant induction of phytotoxins under those conditions (Table 2). This suggests that temperature in a 3 h incubation may have little effect on phytotoxin production for Psa6. To clarify the role of temporal changes in the behavior of virulence genes, such as type III effector genes and phytotoxin synthesis genes, we examined the behavior of gene expression over time using RT-qPCR analysis. Because the relative expression values in RT-qPCR analyses are calculated using constantly expressed reference genes, the selection of appropriate reference genes is important. Based on the conditions described in the Materials and Methods section, we selected five candidates among a total of 5,796 genes (Table S1). From these five genes (50S ribosomal protein L13, bifunctional glutamine synthetase adenylyltransferase/deadenyltransferase, tRNA dihydrouridine (20/20a) synthase *dusA*, transcriptional regulator *ftrA*, and microcin ABC transporter ATP-binding protein), three genes for each bacterium were used as reference genes. For these genes, expression level was high and stable (expression change was low), regardless of the media used. Thus, we designed primers for the RT-qPCR analysis based on the sequences of the following reference genes: 50S ribosomal protein L13, bifunctional glutamine synthetase adenylyltransferase/deadenyltransferase, and tRNA dihydrouridine (20/20a) synthase *dusA* for Psa6; 50S ribosomal protein L13, bifunctional glutamine synthetase adenylyltransferase/deadenyltransferase, and transcriptional regulator *ftrA* for Psa1 and Psa3; and 50S ribosomal protein L13, bifunctional glutamine synthetase adenylyltransferase/deadenyltransferase, and microcin ABC transporter ATP-binding protein for Psg (Table S1). Since these reference genes are a group of genes with stable expression and housekeeping, it is expected that strains other than Psa6 have similar expression tendencies. In addition, it is expected that internal control can be guaranteed by the geometric mean of 3 kinds of genes in order to avoid the bias of expression of each gene.

## Temporal changes in the expression of virulence genes of Psa6 in RT-qPCR analysis

The expression of phytotoxin synthesis genes and type III effector genes of Psa6 was investigated with RT-qPCR. Normalization was performed using the reference genes selected above.

For the reference genes, the amplification efficiency ($E$) of target genes in qPCR was approximately 2.0 (Table S1) and three genes were used as the reference ($n = 3$). Thus, using Eq. (1) in the Materials and Methods section with each $E$ of primer sets, the relative expression levels were calculated from the Ct value at a certain time and the Ct value at 0 h. Here, in order to indicate that the results of RNA-Seq and the results of RT-qPCR did not diverge significantly, the expression levels of the virulence genes in the RT-qPCR analysis for Psa6 and the expression levels of the same genes in the RNA-Seq analysis were compared

Hirose et al. (2020), *PeerJ*, DOI 10.7717/peerj.9697

**Table 2** List of gene expression of Psa6 in difference medium conditions by RNA-Seq analysis (FDR < 0.1 with effect size).

| ID | Product | 27HSC[a] | 18HS[b] | 18HSC[c] | logCPM | PValue | FDR | Effect size |
|---|---|---|---|---|---|---|---|---|
| BUE60_04820 | heat-shock protein | 1.355338 | −2.43875 | −1.31508 | 11.73123 | 2.29E−11 | 1.33E−07 | 0.825882 |
| BUE60_03120 | transposase | 1.020947 | −9.70426 | 0.793676 | 11.37198 | 5.78E−11 | 1.68E−07 | 0.641125 |
| BUE60_14130 | transposase | 0 | 0 | 9.353037 | 8.873037 | 8.51E−10 | 1.64E−06 | 0.59108 |
| BUE60_20545 | peptidase M4 | 2.489317 | 4.443465 | 4.633089 | 9.090263 | 4.33E−08 | 6.26E−05 | 0.843105 |
| BUE60_04770 | hypothetical protein | −0.34373 | −7.61338 | −7.61338 | 8.059493 | 5.40E−08 | 6.26E−05 | 0.670986 |
| BUE60_01870 | phage tail protein | 7.758212 | 6.587112 | 7.860501 | 8.547592 | 3.98E−07 | 0.000384 | 0.541496 |
| BUE60_06970 | hypothetical protein | −2.58953 | −0.8029 | −8.1928 | 8.386577 | 1.29E−06 | 0.001067 | 0.864256 |
| BUE60_14400 | 4a-hydroxytetrahydrobiopterin dehydratase | 7.494481 | 7.602294 | 8.036372 | 8.691885 | 2.15E−06 | 0.001554 | 0.56063 |
| BUE60_02835 | chemotaxis protein | 2.250013 | 1.336789 | 2.235526 | 11.70462 | 2.82E−06 | 0.001818 | 0.626422 |
| BUE60_17905 | DNA-directed DNA polymerase | 0 | 9.167101 | 6.180072 | 8.399849 | 3.17E−06 | 0.001838 | 0.721365 |
| BUE60_02480 | phospholipid-binding protein | 2.75255 | 2.175321 | 2.667653 | 11.94782 | 4.11E−06 | 0.001985 | 0.799186 |
| BUE60_04310 | hypothetical protein | 0.224221 | −7.50516 | −7.50516 | 8.207357 | 3.77E−06 | 0.001985 | 0.637833 |
| BUE60_16610 | type III effector (translocator) HrpK | 0.606558 | 2.038242 | 1.196631 | 12.13895 | 4.71E−06 | 0.002099 | 0.957299 |
| BUE60_04320 | ribosomal subunit interface protein | 1.726174 | 0.111266 | 1.619895 | 11.57393 | 8.20E−06 | 0.003394 | 0.768705 |
| BUE60_28960 | transposase | 0 | 9.007053 | 7.799665 | 8.680074 | 1.35E−05 | 0.004903 | 0.403141 |
| BUE60_03115 | transposase | −0.59594 | 0.589465 | −9.83131 | 10.6735 | 1.30E−05 | 0.004903 | 0.476177 |
| BUE60_23710 | ester cyclase | 0.775814 | −6.16163 | −6.16163 | 7.36076 | 2.09E−05 | 0.006918 | 0.599137 |
| BUE60_21600 | transporter | 0 | 6.717301 | 7.851922 | 7.885333 | 2.15E−05 | 0.006918 | 0.367771 |
| BUE60_05090 | ATP-dependent chaperone ClpB | 1.973675 | −0.24943 | 0.635359 | 9.808781 | 2.36E−05 | 0.007214 | 0.855973 |

Hirose et al. (2020), *PeerJ*, DOI 10.7717/peerj.9697

**Table 2** (*continued*)

| ID | Product | 27HSC[a] | 18HS[b] | 18HSC[c] | logCPM | PValue | FDR | Effect size |
|----|---------|----------|---------|----------|--------|--------|-----|-------------|
| BUE60_16865 | hypothetical protein | 0 | 7.452753 | 0 | 6.847248 | 2.67E−05 | 0.00775 | 0.834247 |
| BUE60_14380 | acetate–CoA ligase | −0.39884 | −3.76361 | −3.20795 | 8.443014 | 3.68E−05 | 0.010001 | 0.773293 |
| BUE60_12810 | Fe/S-dependent 2-methylisocitrate dehydratase AcnD | −2.44697 | −3.73661 | −4.31049 | 7.889943 | 3.80E−05 | 0.010001 | 0.840004 |
| BUE60_09615 | 50S ribosomal protein L11 | 0.386928 | 0.149426 | −6.90695 | 8.113202 | 5.79E−05 | 0.014595 | 0.523026 |
| BUE60_02595 | molecular chaperone DnaK | 1.635999 | −0.18151 | 0.473407 | 11.0735 | 6.45E−05 | 0.015578 | 0.932595 |
| BUE60_02495 | 30S ribosomal protein S15 | −0.67634 | −3.71221 | −3.52621 | 12.0987 | 7.41E−05 | 0.017175 | 0.614689 |
| BUE60_21040 | ribosomal-protein-alanine N-acetyltransferase | 2.254871 | 1.463555 | −5.13538 | 7.73098 | 7.73E−05 | 0.01723 | 0.854178 |
| BUE60_24615 | threonine ammonia-lyase, biosynthetic | −2.31336 | −7.75066 | −2.51696 | 7.85215 | 8.26E−05 | 0.017737 | 0.580708 |
| BUE60_14740 | hypothetical protein | −6.33963 | −6.33963 | 0.83639 | 7.541523 | 8.97E−05 | 0.018564 | 0.670119 |
| BUE60_19565 | hypothetical protein | 1.419291 | 2.063655 | 2.243516 | 10.73711 | 0.000108 | 0.021543 | 0.754757 |
| BUE60_17855 | hypothetical protein | 1.644416 | 0.041077 | 0.333665 | 10.61235 | 0.000112 | 0.02158 | 0.934538 |
| BUE60_15390 | gamma carbonic anhydrase family protein | 0.672182 | −0.73084 | −6.19487 | 7.496001 | 0.000115 | 0.02158 | 0.722079 |
| BUE60_02340 | transposase | −5.51749 | 3.725977 | 1.970766 | 8.817886 | 0.000128 | 0.023196 | 0.714283 |
| BUE60_12815 | 2-methylcitrate synthase | −3.05674 | −7.53602 | −3.3801 | 7.531794 | 0.000132 | 0.023209 | 0.872486 |
| BUE60_18535 | ribosomal subunit interface protein | 1.422023 | −0.98943 | 0.057456 | 10.76842 | 0.000151 | 0.025747 | 0.757798 |
| BUE60_19430 | hypothetical protein | 1.211633 | −5.7711 | −5.7711 | 7.283092 | 0.000158 | 0.0261 | 0.73941 |
| BUE60_13595 | hypothetical protein | 1.636492 | −5.54256 | −5.54256 | 7.34782 | 0.000186 | 0.027557 | 0.412538 |
| BUE60_21160 | hypothetical protein | 0.130148 | −6.04703 | −6.04703 | 6.997812 | 0.00019 | 0.027557 | 0.777151 |
| BUE60_19345 | glutathione S-transferase | 0 | 6.553643 | 7.982156 | 7.93946 | 0.000185 | 0.027557 | 0.497624 |
| BUE60_15180 | hypothetical protein | −0.64369 | −6.95686 | −6.95686 | 7.415176 | 0.000178 | 0.027557 | 0.528899 |
| BUE60_07695 | Fis family transcriptional regulator | −6.07158 | 0.083934 | −6.07158 | 6.817511 | 0.000184 | 0.027557 | 0.867404 |
| BUE60_24965 | hypothetical protein | 1.091995 | −5.7761 | −5.7761 | 7.223623 | 0.000198 | 0.027965 | 0.76029 |

Hirose et al. (2020), *PeerJ*, DOI 10.7717/peerj.9697

**Table 2** (*continued*)

| ID | Product | 27HSC[a] | 18HS[b] | 18HSC[c] | logCPM | PValue | FDR | Effect size |
|---|---|---|---|---|---|---|---|---|
| BUE60_21340 | hypothetical protein | 0.489961 | −0.87607 | −6.23156 | 7.435516 | 0.000214 | 0.029594 | 0.678285 |
| BUE60_01650 | thiosulfate sulfurtransferase | −6.07151 | 0.083998 | −6.07151 | 6.817511 | 0.000234 | 0.031502 | 0.867404 |
| BUE60_10700 | molecular chaperone DnaK | 0 | 6.590735 | 7.732228 | 7.78859 | 0.000254 | 0.033474 | 0.499649 |
| BUE60_27730 | glycoside hydrolase 68 family protein | 1.990119 | 1.550896 | 1.567534 | 10.9663 | 0.000266 | 0.034293 | 0.908225 |
| BUE60_02230 | prevent-host-death protein | 0.759926 | −5.94885 | −5.94885 | 7.190749 | 0.000284 | 0.035773 | 0.772524 |
| BUE60_17400 | transposase | −6.7498 | −0.97421 | −6.7498 | 7.043704 | 0.0004 | 0.049323 | 0.710718 |
| BUE60_09700 | 30S ribosomal protein S3 | −0.25804 | −0.86626 | −1.83476 | 10.5992 | 0.000433 | 0.052335 | 0.756287 |
| BUE60_25785 | chaperonin GroL | 0.57601 | −1.49145 | −0.64014 | 10.64873 | 0.000447 | 0.052893 | 0.806498 |
| BUE60_23685 | type III chaperone protein ShcN | 0.791859 | 2.274048 | 1.261047 | 10.05967 | 0.0005 | 0.057302 | 0.729401 |
| BUE60_06660 | hypothetical protein | 0.308284 | 0.295896 | −5.96605 | 7.341313 | 0.000504 | 0.057302 | 0.772319 |
| BUE60_12330 | cysteine hydrolase | 2.283692 | 2.685102 | −4.08027 | 7.224962 | 0.000553 | 0.060751 | 0.598509 |
| BUE60_18905 | HslU–HslV peptidase proteolytic subunit | 1.65815 | 0.192043 | 2.487488 | 9.847487 | 0.000566 | 0.060751 | 0.826576 |
| BUE60_28885 | IS66 family transposase | 0.407687 | −6.25623 | −6.25623 | 7.261808 | 0.000566 | 0.060751 | 0.430101 |
| BUE60_26610 | transposase | 0 | 7.163183 | 8.603967 | 8.477509 | 0.000609 | 0.064132 | 0.24148 |
| BUE60_19335 | hypothetical protein | 3.39819 | 1.071081 | −4.22353 | 7.607979 | 0.000658 | 0.065776 | 0.780608 |
| BUE60_17470 | 1-(5-phosphoribosyl)-5-((5-phosphoribosylamino)methylideneamino)imidazole-4- carboxamide isomerase | 0.1607 | −6.29092 | −6.29092 | 7.188882 | 0.000652 | 0.065776 | 0.404107 |

Hirose et al. (2020), *PeerJ*, DOI 10.7717/peerj.9697

**Table 2** (*continued*)

| ID | Product | 27HSC[a] | 18HS[b] | 18HSC[c] | logCPM | PValue | FDR | Effect size |
|---|---|---|---|---|---|---|---|---|
| BUE60_12825 | GntR family transcriptional regulator | −3.71429 | −7.56884 | −3.23523 | 7.521741 | 0.000643 | 0.065776 | 0.622523 |
| BUE60_29200 | hypothetical protein | −2.68519 | −7.0603 | −7.0603 | 7.110864 | 0.000695 | 0.068254 | 0.751303 |
| BUE60_07150 | IS3 family transposase | 3.570854 | −3.93558 | −3.93558 | 7.380453 | 0.000707 | 0.068298 | 0.528357 |
| BUE60_21075 | phna protein alkylphosphonate uptake | 0.018146 | −1.62145 | −6.47975 | 7.385461 | 0.000726 | 0.068957 | 0.759865 |
| BUE60_27745 | hypothetical protein | 1.89857 | 0.396512 | −6.92905 | 8.922955 | 0.000745 | 0.069619 | 0.644543 |
| BUE60_21475 | ABC transporter substrate-binding protein | 2.601267 | 0.065625 | 0.341459 | 8.703373 | 0.000796 | 0.073259 | 0.828218 |
| BUE60_28550 | hypothetical protein | 6.282644 | 8.43938 | 0 | 7.912608 | 0.000845 | 0.075941 | 0.40412 |
| BUE60_15495 | hypothetical protein | 1.808798 | 2.130559 | 1.632545 | 9.785202 | 0.000852 | 0.075941 | 0.825144 |
| BUE60_22845 | hypothetical protein | −6.87515 | 1.564223 | −6.87515 | 8.061091 | 0.000944 | 0.082881 | 0.365741 |
| BUE60_02805 | D-hexose-6-phosphate mutarotase | −0.98576 | −3.05392 | −1.17007 | 9.230321 | 0.000979 | 0.084697 | 0.784073 |
| BUE60_28540 | phospholipase | 0 | 7.511676 | 0 | 6.884314 | 0.001001 | 0.085306 | 0.45 |
| BUE60_15770 | dipeptide ABC transporter permease DppC | −3.81395 | 3.113897 | −3.81395 | 6.646059 | 0.001023 | 0.085963 | 0.866672 |
| BUE60_25850 | IS630 family transposase | −5.25668 | 2.859024 | −5.25668 | 7.527171 | 0.001048 | 0.08674 | 0.421919 |
| BUE60_23415 | hypothetical protein | −6.49249 | −0.63362 | −6.49249 | 6.929545 | 0.001133 | 0.092485 | 0.629163 |
| BUE60_06270 | efflux transporter periplasmic adaptor subunit | −1.16271 | −0.59189 | −6.46943 | 7.196219 | 0.00117 | 0.092932 | 0.674268 |
| BUE60_02790 | sugar ABC transporter permease | −1.60463 | −1.04277 | −1.66657 | 10.4278 | 0.001167 | 0.092932 | 0.729736 |
| BUE60_11475 | Cd(II)/Pb(II)-responsive transcriptional regulator | 1.789462 | 1.420796 | −4.80667 | 7.242217 | 0.001252 | 0.098092 | 0.730896 |
| BUE60_08690 | flagellar biosynthetic protein FliO | 1.326382 | −5.15132 | −5.15132 | 6.890662 | 0.001273 | 0.098371 | 0.849329 |
| BUE60_28090 | transposase | 0 | 7.452437 | 0 | 6.844453 | 0.001307 | 0.098371 | 0.45 |
| BUE60_19630 | hypothetical protein | −2.56629 | −6.85827 | −6.85827 | 6.97748 | 0.001307 | 0.098371 | 0.925232 |
| BUE60_01880 | phage tail protein | 6.780487 | 5.307834 | 6.57077 | 7.564627 | 0.001339 | 0.099517 | 0.635214 |

**Notes.**

[a] Gene expression ratio in 27 °C-HSC medium against 27 °C-HS medium.

[b] Gene expression ratio in 18 °C-HS medium against 27 °C-HS medium.

[c] Gene expression ratio in 18 °C-HSC medium against 27 °C-HS medium.

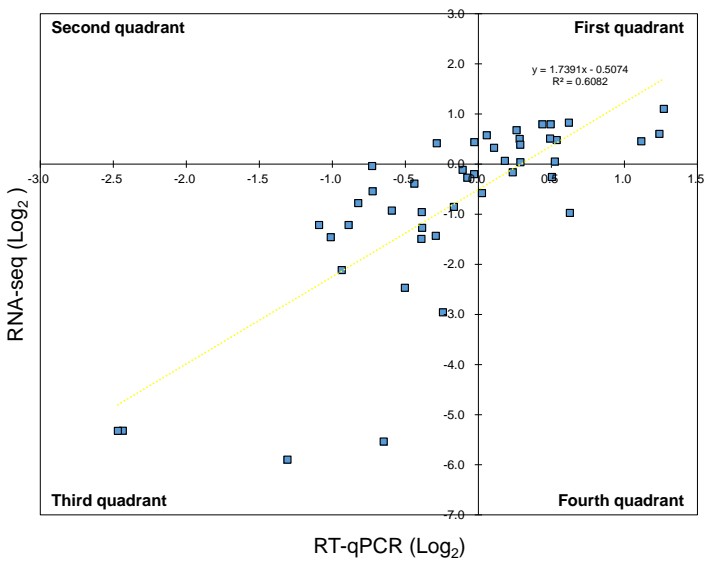

**Figure 1 Comparison of expression trends in the RNA-seq and RT-qPCR analyses.** The expression levels of the virulence genes in RNA-Seq were plotted on the vertical axis and that in RT-qPCR were plotted on the horizontal axis. Most of the plots were concentrated in the first and third quadrants, while some were concentrated in the second and fourth quadrants. The yellow wavy line indicates a linear approximation, and $R^2$ value (about 0.6) was obtained.

(Table S4). When the expression levels of the virulence genes from RNA-Seq were plotted on the vertical axis and those from RT-qPCR were plotted on the horizontal axis, both RNA-Seq and RT-qPCR results were correlated (Fig. 1). The plots were concentrated in the first and third quadrants ($R^2$ (coefficient of determination) value is about 0.6), although some genes were expressed differently. We concluded that the results from RT-qPCR can almost function as a representation of transcriptome analysis.

The expression of phytotoxin synthesis genes and effector-related genes of Psa6 under various culture conditions with different incubation times was investigated using RT-qPCR (Table S5 & Data S1). From the results of RT-qPCR analysis, heat maps were output (Fig. 2). We found that phytotoxin synthesis genes and effector genes of Psa6 were induced in HS and HSC media (Fig. 2A), which are known to induce coronatine production. In contrast, these genes were not induced in the *hrp*-inducing medium, which is known to induce *hrp*-dependent effector genes. The LB and *hrp*-inducing media were clustered together and were separate from the HS and HSC media. In the LB and *hrp*-inducing media cluster, most genes were suppressed. Moreover, in the *hrp*-inducing medium, no clear induction of the *hrp* L-dependent effector genes of Psa6 was observed. On the other hand, phytotoxin synthesis genes and effector genes were significantly induced in HS and HSC media. Additionally, incubation at 18 °C and 27 °C did not result in significant changes to gene expression; samples from each temperature were clustered separately. This suggests that culture medium composition and incubation time have a stronger effect on Psa6 cultivation than incubation temperature.

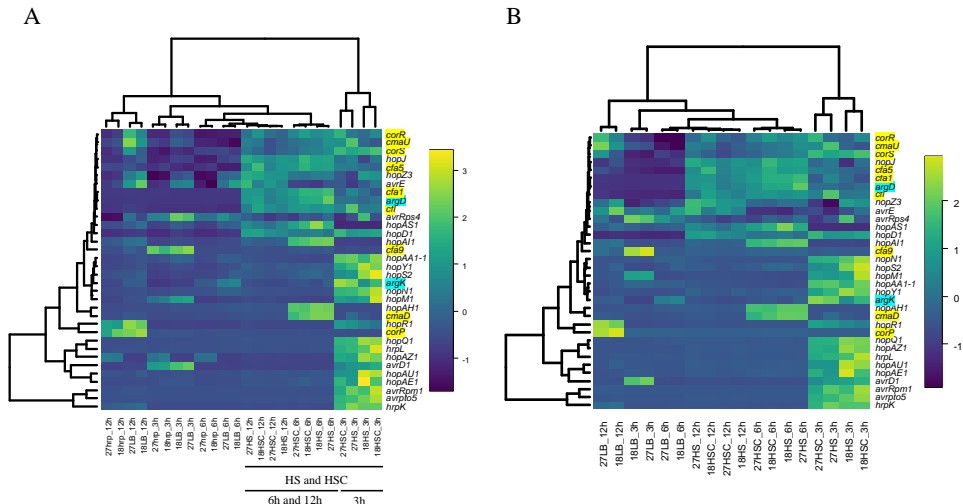

**Figure 2  Heat map constructed from the expression level of virulence genes of Psa6 under each culture condition.** The results of RT-qPCR analysis of Psa6 were clustered using the unweighted pair group method with arithmetic mean (UPGMA). (A) Heat map of expression levels of virulence genes under each culture condition. (B) Reconstructed heat map excluding the conditions of the *hrp*-inducing medium. Gene names highlighted in blue are phaseolotoxin synthesis genes, and gene names highlighted in yellow are coronatine synthesis genes. Remaining genes are type III effectors. In heat maps, induced genes are indicated as yellow boxes and suppressed genes are indicated as blue boxes.

Intriguingly, the expression of virulence genes was generally separated into the following two groups: early-inducible genes and late-inducible genes (Fig. 2). This was more clearly observed in the heat map, excluding the conditions of *hrp*-inducing medium (Fig. 2B). The early-inducible genes (*hrpL*, *avrD1*, *avrpto5*, *avrRpm1*, *hopAA1-1*, *hopAE1*, *hopAU1*, *hopAZ1*, *hopM1*, *hopN1*, *hopQ1*, *hopR1*, *hopS2*, *hopY1*, *hrpK,* and *argK* ) were induced after 3 h of incubation but suppressed at 6 and 12 h. Most of them were thought to be *hrpL*-dependent effector genes (*Lam et al., 2014*; *Vencato et al., 2006*). The early-inducible genes of Psa6 contains *hrpL* and several effector genes. It is known that the HrpL protein binds to the Hrp box promoter sequence on the upper region of protein-encoding sequences in several effector genes (*Greenberg & Yao, 2004*; *Lam et al., 2014*), then, we could also find the Hrp box promoter sequences in some early-inducible genes of Psa6 (Fig. S2). Therefore, HrpL regulation may be involved in the early-inducible genes of Psa6 in response to HS and HSC media. However, further research is needed on this detail. In contrast, the late-inducible genes (*avrE1*, *avrRps4*, *hopAH1*, *hopAI1*, *hopAS1*, *hopD1*, *hopJ*, *hopZ3*, *cfl*, *cfa1*, *cfa5*, *cfa9*, *cmaD*, *cmaU*, *corP*, *corR*, *corS,* and *argD*) were induced after 6 h of incubation. These genes included phytotoxin synthesis genes, except *argK* and some effector genes. Thus, when Psa6 is incubated under phytotoxin-induced conditions (HS and HSC media), induction of phytotoxin synthesis genes followed that of effector genes. In addition, phytotoxin synthesis genes were expressed for longer than the *hrpL*-regulated effectors.

## Comparison of gene expression behavior between Psa6 and other bacteria

As described above, in Psa6, the effector genes are not induced in the *hrp*-inducing medium, and the induction of virulence genes in phytotoxin-inducing media is strictly controlled over time. It was unclear whether these behaviors of gene expression were Psa6 specific; thus, similar tests were performed on the other bacteria. The expressions of virulence genes in two biovars (Psa1 and Psa3) and one other species (Psg) were investigated using RT-qPCR (Table S6 & Data S2). Normalization was performed using the reference genes selected above. The expression level of virulence genes was compared across all bacteria. Heat maps were generated from the expression patterns of the effector genes shared by each bacterium (Fig. 3). Psa1 and Psa3 were nested in the heat map (Fig. 3A). In Psa6, the expression of effector genes was significantly induced compared with other bacteria. Furthermore, Psg was scattered in clusters of Psa1 and Psa3, and the clusters did not split due to differences in pathovars. When clustering was performed with only common effectors of both Psa1 and Psa3 (omitting Psa6 and Psg), medium composition had no effect on gene expression (Fig. 3B). Our results show that medium composition and incubation time did not significantly affect the expression of effector genes in Psa1, Psa3, or Psg.

We also compared phytotoxin synthesis genes of Psa6 with other bacteria. To compare the expression of coronatine synthesis genes in Psa6 and Psg, the expression levels of *cfa1* and *cmaD* at 0, 3, and 6 h incubation were examined (Table 3). The results showed that incubation of Psa6 in HS and HSC media induced the expression of *cfa1* and *cmaD* over time, but incubation in LB and *hrp*-inducing media significantly suppressed *cfa1* and slowly induced the expression of *cmaD*. Therefore, the behavior of the coronatine synthesis genes of Psa6 was affected by the medium composition. On the other hand, Psg was not affected by medium composition. In Psg, there were no differences in gene expression across the different media, and only *cfa1* was induced over time regardless of medium composition. There was no induction or suppression of *cmaD*. Additionally, we compared the expression of phaseolotoxin synthesis-related genes in Psa6 and Psa1 (Table 4). In Psa6, the expression of *argK* and *argD* was slowly induced by culture in HS and HSC media. In addition, the expression of *argK* was notably induced and *argD* was greatly suppressed in LB and *hrp*-inducing media. However, in Psa1, *argK* was induced in LB and *hrp*-inducing media, and *argD* remained expressed at a constant level without induction or suppression. In summary, the coronatine and phaseolotoxin synthesis genes in Psa6 differed in their response to medium composition and incubation time compared to other bacteria, with a particularly strong effect of medium composition.

## Expression patterns and conditions of the virulence genes in Psa6

Coronatine production in Psg PG 4180 is more active in cultures at 18 °C than 24 °C or higher (*Palmer & Bender, 1993*). In contrast, coronatine production in *P. syringae* pv. *tomato* (Pto) DC 3000 is temperature independent and constant (*Braun et al., 2008*). Additionally, seven strains of *P. syringae*, including Psg PG 4180, Psg 7a, and Pto PT 23.2, produce more coronatine in HSC medium than in HS medium (*Palmer & Bender, 1993*). However, two strains of *P. syringae* pv. *maculicola* 438 and Pst DC 3000 do not produce
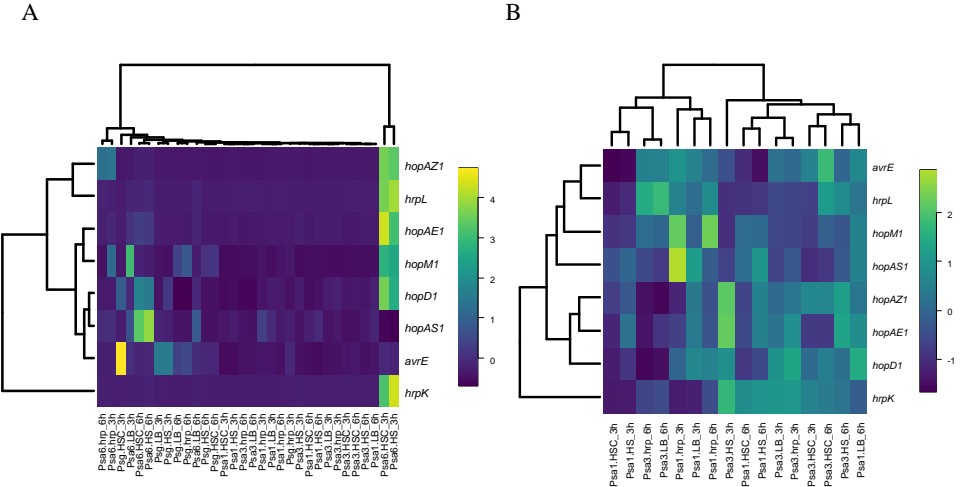

A                                                                 B

**Figure 3    Heat map constructed from the expression levels of virulence genes of Psa6, Psa1, Psa3, and Psg under each culture condition.** The results of RT-qPCR analysis of Psa6, Psa1, Psa3, and Psg were clustered using the unweighted pair group method with arithmetic mean (UPGMA). (A) Heat map of expression levels of virulence genes under each culture condition. (B) Reconstructed heat map excluding the results of both Psa6 and Psg. In heat maps, induced genes are indicated as yellow boxes and suppressed genes are indicated as blue boxes.

**Table 3    Expression of *cfa1* and *cmaD* of Psa6 and Psg by RT-qPCR analysis.**

| Gene | Condition | 0 h | 3 h | 6 h | Gene | Condition | 0 h | 3 h | 6 h |
|------|-----------|-----|-----|-----|------|-----------|-----|-----|-----|
| Psa6 *cfa1* | 27HS | 0.00 | **1.87** | **2.83** | Psg *cfa1* | 27HS | 0.00 | 0.08 | **3.84** |
|  | 27HSC | 0.00 | 0.79 | **2.43** |  | 27HSC | 0.00 | 0.27 | **3.76** |
|  | 27LB | 0.00 | −3.90 | −4.55 |  | 27LB | 0.00 | 0.58 | **3.73** |
|  | 27*hrp* | 0.00 | −3.46 | −3.88 |  | 27*hrp* | 0.00 | 0.29 | **3.55** |
| Psa6 *cmaD* | 27HS | 0.00 | **3.72** | **5.73** | Psg *cmaD* | 27HS | 0.00 | 0.18 | −0.29 |
|  | 27HSC | 0.00 | **1.32** | **5.41** |  | 27HSC | 0.00 | 0.57 | 0.06 |
|  | 27LB | 0.00 | 0.89 | **1.30** |  | 27LB | 0.00 | 0.86 | −0.58 |
|  | 27*hrp* | 0.00 | 0.78 | **1.40** |  | 27*hrp* | 0.00 | 0.57 | −0.56 |

**Notes.**
The expression level (log2) is indicated as a ratio relatively against 0h of each culture condition. Bold letters indicate that expression is induced (log2 > 1.0), and blue letters indicate that expression is suppressed (log2 < −1.0).

coronatine in HSC medium (*Palmer & Bender, 1993*). Moreover, in Psg PG 4180, the CmaB protein, a synthesis protein of coronatine, decreases 4 h after the temperature is changed from 18 °C to 28 °C, suggesting that changes in gene expression can be observed at an earlier stage (*Budde et al., 1998*). However, the coronatine synthesis genes of Psa6 were induced later, regardless of temperature (Fig. 2). Thus, the coronatine synthesis genes of Psa6 may be regulated by different environmental conditions than Psg PG 4180. In addition, in the strain of Psg used in this study (MAFF 301683), which was isolated in Japan, the expression of the coronatine synthesis genes was induced regardless of medium

**Table 4** Expression of *argK* and *argD* of Psa6 and Psa1 by RT-qPCR analysis.

| Gene | Sample | 0 h | 3 h | 6 h | Gene | Sample | 0 h | 3 h | 6 h |
|---|---|---|---|---|---|---|---|---|---|
| Psa6 *argK* | 27HS | 0.00 | **4.81** | **3.13** | Psa1 *argK* | 27HS | 0.00 | 0.28 | **1.78** |
| | 27HSC | 0.00 | **4.93** | **2.66** | | 27HSC | 0.00 | −0.24 | 0.89 |
| | 27LB | 0.00 | **1.85** | **3.78** | | 27LB | 0.00 | **2.09** | 0.85 |
| | 27*hrp* | 0.00 | **1.31** | **2.92** | | 27*hrp* | 0.00 | **1.97** | 0.95 |
| Psa6 *argD* | 27HS | 0.00 | 0.89 | **2.74** | Psa1 *argD* | 27HS | 0.00 | −0.30 | 0.12 |
| | 27HSC | 0.00 | 0.91 | **2.04** | | 27HSC | 0.00 | −0.71 | −0.35 |
| | 27LB | 0.00 | *−5.12* | *−5.20* | | 27LB | 0.00 | −0.01 | 0.30 |
| | 27*hrp* | 0.00 | *−5.39* | *−5.29* | | 27*hrp* | 0.00 | −0.01 | 0.03 |

Notes.

The expression level (log2) is indicated as a ratio relatively against 0h of each culture condition. Bold letters indicate that expression is induced (log2 > 1.0), and blue letters indicate that expression is suppressed (log2 < −1.0).

composition (Table 3). Conversely, in a different Psg strain, PG 4180, coronatine synthesis was induced in HS and HSC media (*Palmer & Bender, 1993*). Thus, depending on the type of *Pseudomonas* (pathovars and strains), coronatine production is thought to be affected by various environmental conditions.

Moreover, despite Psa6 and Psa1 possessing a similar phaseolotoxin synthesis gene cluster (*Fujikawa & Sawada, 2019*), both expressed them differently (Table 4). This gene cluster is contained in the *tox* island in the genomes of Psa6 and Psa1. However, in Psa6, the *ginABCD* operon, which is located at the left end of the *tox* island and is involved in the excision/insertion of the *tox* island, was truncated, suggesting that the island of biovar 6 might have lost its mobility (*Fujikawa & Sawada, 2019*). At present, it is unknown whether the expression pattern of the phaseolotoxin synthesis genes is different due to differences in the *ginABCD* operon or whether it depends on another regulation system. This requires further investigation.

We found that Psa6 did not induce *hrpL*-dependent effectors in the *hrp*-inducing medium but was induced in HS and HSC media (Fig. 2). The *hrp*-inducing medium is an oligotrophic medium (*Huynh, Dahlbeck & Staskawicz, 1989*), but HS and HSC media contain rich sugars and minerals (*Palmer & Bender, 1993*). In Psa6, phytotoxin production and effector-related genes were induced similarly even with different concentrations of glucose (10 g/L in HS medium and 20 g/L in HSC medium) (Table 1). However, fructose (1.8 g/L) in the *hrp*- inducing medium was not significantly induced. This may be due to qualitative or quantitative differences in carbon sources. Additionally, it seemed that the FeCl$_3$ concentration (HS, HSC, and the *hrp*-inducing media) (Table 1) may not be involved in the expression behavior of phytotoxin synthesis genes and effector genes. In this study, HS and HSC media were found to induce the virulence genes of Psa6. These genes were divided into two groups according to how long it took for them to be induced.

The early-inducible genes included many effectors and *hrpL*. The effectors in this group are involved in plant immunity such as pathogen-associated molecular patterns (PAMPs) triggered immunity (PTI) and effector-triggered immunity (ETI) (*Jones & Dangl, 2006*).

PTI is a plant immune system that responds to the cellular components carried by pathogens such as flagella protein and elongation factor. A plant immune system activated by PTI induces various antimicrobial molecules such as salicylic acid and PR (pathogenesis-related) proteins, resulting in the inhibition of pathogen spread. In response, pathogens can interfere with PTI by injecting the effectors to avoid the plant defense. However, plants have ETI, another plant immune system, which responds to the effectors. Thus, the plant immune system is expressed as a zigzag model (*Jones & Dangl, 2006*). Although the function of these effectors in Psa6 to the host plants is unknown, these homologous proteins have been reported as follows; AvrPto5, contained in the early-inducible gene of Psa6, induces PTI, whereas HopS, HopAA1, and HopM1 inhibit PTI. In addition, AvrPto5, AvrRpm1, and HopD induce ETI, whereas HopD, HopN, and HopS suppress ETI (*Lindeberg, Cunnac & Collmer, 2012*).

In contrast to the early-inducible genes, the late-inducible genes included phytotoxin synthesis-related genes other than *argK*. Because it has been reported that coronatine and other phytotoxins accumulate over time (*Palmer & Bender, 1993*), phytotoxin synthesis-related genes may require longer periods than most effector genes for expression. On the other hand, the *argK* gene of Psa6 was an early inducible gene. Because *argK* is also known to be involved in the production of phaseolotoxin-insensitive ornithine carbamoyltransferase (*Sawada, Takeuchi & Matsuda, 1997*), this early expression may lead to resistance to phaseolotoxin and subsequent phaseolotoxin production. Additionally, some effector genes were listed in the late-inducible genes. It is known that AvrE and AvrRps4 in Pto suppress PTI (*Lindeberg, Cunnac & Collmer, 2012*), while HopZ3, AvrRps4, HopAS1, and HopAI1 induce ETI (*Lewis et al., 2014*; *Lindeberg, Cunnac & Collmer, 2012*; *Sohn, Zhang & Jones, 2009*; *Sohn et al., 2012*), and HopZ suppresses ETI in *Arabidopsis* (*Lindeberg, Cunnac & Collmer, 2012*). These late-inducible genes in Psa6 may also trigger ETI, such as the Pto-*Arabidopsis* interaction. However, it is thought to function under conditions different from those of the early-inducible genes, for example, at different times, in different environment, and with different hosts.

The coronatine synthesis genes of Psa6 are located on plasmids whose sequences are highly homologous to those of Psa2 and Psg (*Fujikawa & Sawada, 2019*). The phaseolotoxin production genes of Psa6 are located on the *tox* island in the genome, and it is thought that the region was originally in another bacterium and transferred to Psa1 and Psa6 through horizontal gene transfer (*Sawada, 2016*; *Fujikawa & Sawada, 2019*). In addition, along with the presence or absence of virulence factors, it was suggested that the regulation of gene expression varied also among pathovars and biovars. Phytopathogenic bacteria are then thought to have coevolved with plants, with their behavior and infection method related to the life cycle of the host plant. Psa6, with two phytotoxins, was unique not only in its physiological characteristics but also in the expression behavior of phytotoxin synthesis genes and effector genes. Although the relationship between the number of virulence factors and their expression behavior is still not clear, we propose that strict gene expression is thought to have been acquired with the acquisition of many virulence factors. The behavior of virulence genes in Psa clarified in this study will help elucidate the differences in virulence and survival strategies among biovars.

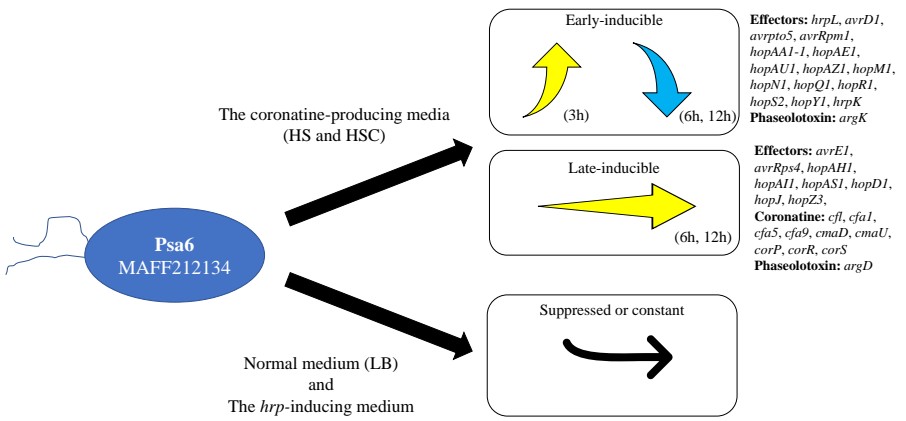

**Figure 4** **Schematic diagram of the expression pattern of Psa6 virulence genes in different media.** The expression of virulence genes (type III effector genes and phytotoxin synthesis genes) of Psa6 was induced in the coronatine-producing media, and the expression behavior had two patterns: early-inducible and late-inducible. These genes were suppressed or constant in LB medium and the *hrp*-inducing medium.

## CONCLUSIONS

*Pseudomonas syringae* pv. *actinidiae* biovar 6 (Psa6) is a causal agent of kiwifruit bacterial canker and is a unique plant pathogenic bacterium, producing two types of phytotoxins, coronatine and phaseolotoxin. We investigated the expression behavior of virulent genes of Psa6 under various culture conditions. The expression pattern of phytotoxin synthesis genes in Psa6 was different from that of another pathovar, *P. s.* pv. *glycinea*, which produces coronatine, and another biovar, Psa1, which produces phaseolotoxin. In addition, the expression of virulence genes (phytotoxin synthesis genes and type III effector genes) was induced in the coronatine-producing media regardless of temperature, and confirmed that the expression behavior had two patterns: early-inducible and late-inducible (Fig. 4). The expression behavior of virulence genes of Psa6 clarified in this study will contribute to elucidating the virulence of Psa to kiwifruits and survival strategies among Psa biovars and plant pathogenic *Pseudomonas*.

## ACKNOWLEDGEMENTS

We are grateful to Dr. Takako Ishiga, Ms. Mariko Taguchi, Ms. Akane Sasaki, and Ms. Hiroe Hatomi for their support with the experiments. We also thank the members of the Laboratory of Plant Parasitic Mycology, University of Tsukuba, and the members of IFTS-NARO for their helpful discussions. We would like to thank Editage for English language editing.

### Funding
The authors received no funding for this work.

### Competing Interests
The authors declare there are no competing interests.

### Author Contributions
- Karin Hirose and Takashi Fujikawa conceived and designed the experiments, performed the experiments, analyzed the data, prepared figures and/or tables, authored or reviewed drafts of the paper, and approved the final draft.
- Yasuhiro Ishiga conceived and designed the experiments, analyzed the data, authored or reviewed drafts of the paper, and approved the final draft.

### Data Availability
  RNA-Seq data including raw data are available at NCBI GEO: GSE149743.
  RT-qPCR data including raw data are available in Supplementary Files.

### Supplemental Information
Supplemental information for this article can be found online at http://dx.doi.org/10.7717/peerj.9697#supplemental-information.

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
