# Peer review of "Phytotoxin synthesis genes and type III effector genes of Pseudomonas syringae pv. actinidiae biovar 6 are regulated by culture conditions"

_PeerJ, doi:10.7717/peerj.9697_

## Round 0.1 · original submission · Major Revisions

This manuscript has been evaluated by two experts. While the reviewers were encouraging of the themes of the article, both also made a number of good suggestions as to how to improve the article prior to publication.

In particular, for your resubmission, please pay attention to both reviewers comments about statistical tests and adequate controls for these experiments. I think there are ways given your data to avoid an extensive amount of new experiments, but it would be good to provide greater context for the nuances of interpreting your results. Please do take note of reviewer 2's extensive comments on experimental design during the resubmission though.

Reviewer 1 ·

Basic reporting

no comment

Experimental design

I would like to see a statistical test applied to the results of Figure 1 to make its findings more robust.

Validity of the findings

well done, only further clarification needed.

Additional comments

line 124-128: Please provide version #s for all bioinformatic processing software
line 174: I presume size screening was also part of the confirmation of PCR products.

Line 237: For figure 1, I anticipated a figure that showed the degree of fit in the correlation between the RNA-seq and RT-PCR analysis. I would encourage you to change the way this is plotted to increase the confidence of fit.

Line 259 - "most of them were thought to be hrpL-dependent effector genes" needs a citation.

Paragraph starting line 266: This section is a bit difficult to follow. I recommend you pull parts out it and incorporate that into the earlier paragraph. Tie the lack of Hrp box sequences to the longer-expressed phytotoxin synthesis genes

I appreciate the many supplemental figures the authors provide. I do wonder if perhaps a final main figure for the text could be a cartoon that shows the expression pathway and comparative expression they discuss towards the end of the paper.

Reviewer 2 ·

Basic reporting

The language is acceptable although usage of "it is thought" as a default statement for well verified results could be substituted with stronger language

Raw RNAseq reads do not appear to be archived in a public database.

This claim that Psa6 is the only P syringae strain that produces two phytotoxins is not true.
Most Psy pv. syringae strains produce three toxins syringomycin,syringopeptin, and syringolin. Some P syringae pv syringae strains produce phaseolotoxin in addition to the toxin triad and some also produce mangotoxin
https://doi.org/10.1371/journal.pone.0036709.
Pto DC3000 produces both coronatine and phevamine A.

Considering the importance of culture medium to this study, the composition per liter for each medium should be included in the methods at least. A table comparing the four media would be preferable.

Be more specific about "using R software", which packages?

Consider converting red/green heatmaps to blue/yellow to accommodate the color blind.

In Figure 1, include the R^2 value.

Experimental design

There are several problems in the design of the RT-qPCR experiment

The PCR efficiency of test genes hasn't been conducted.

In addition you should calculate NRQ using empirically determined efficiencies rather than just assuming the efficiency to be perfect (=2).

Melt curve analysis doesn't seem to have been conducted to verify the amplification of a single product under actual testing conditions.

It isn't clear to me if the primers designed for testing expression of Psa6 genes were verified as valid for the other strains. Different reference genes were used for each strain?

Ideally reference genes should be validated for use and several levels of expression so that test genes can be compared to a references with similar levels of expression. Using all high expression reference genes could result in artifacts and bias. I recognize that there are different opinions on use of reference gene. There should be some editorial guidance on whether this is an issue worth addressing.

Validity of the findings

Other Psy strains were included to verify regulation differences from Psa6 but I really would have been better to use a well validated strain like Pto DC3000 known to respond well to HMM. This would provide a good internal control for the study.

Line 273, The claim the late induced effectors lack hrp box promoters needs a deeper analysis. Are theses genes in operons downstream of other hrp regulated genes?

Line 353- citation that salicylic acid is antimicrobial? chitnases are not likely to exert an effect on bacteria.

The discussion of specific effectors doesn't bring much value. There is not obvious trend identified in either group and each effector is likely to exert these roles only in the context of specific hosts.

---

## Round 0.2 · accepted · Accept

Thank you for your resubmission to PeerJ. The reviewers critiques were both addressed, and I'm happy to accept this for publication.

Reviewer 1 ·

Basic reporting

all corrections were made

Experimental design

thank you for adding the R2

Validity of the findings

All corrections were made

Reviewer 2 ·

Basic reporting

no comment

Experimental design

no comment

Validity of the findings

no comment

Additional comments

My previous concerns have been adequately addressed. Thank you.